# Demonstration of a Learning-Empowered Fiber Specklegram Sensor Based on Focused Ion Beam Milling for Refractive Index Sensing

**DOI:** 10.3390/nano13040768

**Published:** 2023-02-18

**Authors:** Liangliang Gu, Han Gao, Haifeng Hu

**Affiliations:** 1School of Optical-Electrical and Computer Engineering, University of Shanghai for Science and Technology, Shanghai 200093, China; 2Zhangjiang Laboratory, 100 Haike Road, Shanghai 201204, China; 3Institute of Modern Optics, Nankai University, Tianjin 300350, China

**Keywords:** optical fiber sensor, refractive index sensor, focused ion beam (FIB) milling

## Abstract

We report a simple and robust fiber specklegram refractive index sensor with a multimode fiber-single mode fiber-multimode fiber structure based on focused ion beam milling. In this work, a series of fluid channels are etched on the single-mode fiber by using focused ion beam milling to enhance the interaction between light and matter, and a deep learning model is employed to demodulate the sensing signal according to the speckle patterns collected from the output end of the multimode fiber. The feasibility and effectiveness of the proposed scheme were verified by rigorous experiments, and the test results showed that the demodulation accuracy and speed could reach 99.68% and 4.5 ms per frame, respectively, for the refractive index range of 1.3326 to 1.3679. The proposed sensing scheme has the advantages of low cost, easy implementation, and a simple measurement system, and it is expected to find applications in various chemical and biological sensing.

## 1. Introduction

Fiber optic sensors have shown extraordinary potential in sensing applications such as chemical sensing, structural health detection, and biological and biomedical sensing due to their small size, high sensitivity, and immunity to electromagnetic interference [1,2,3,4,5]. Among them, refractive index sensing is one of the most fundamental and critical sensing applications of fiber optic sensors, which can be implemented using devices such as Fabry-Perot interferometers [6,7], long period gratings (LPG) [8,9], fiber Bragg gratings (FBG) [10,11], photonic crystal fibers (PCF) [12], and Mach–Zehnder interferometers [13]. These optical fiber sensors are being intensively explored for sensing the refractive index of fluid and have achieved remarkable results. Unfortunately, most of these methods rely on expensive sensor interrogation schemes, resulting in their applicability perhaps being limited by different factors such as cost or repeatability [14]. In this context, fiber specklegram sensors (FSS), which have high sensitivity and require only relatively simple measurement devices, stand out as promising alternatives to the widely used fiber optic sensing technology, and they have been applied in various sensing applications [15,16,17].

Various techniques have been reported for demodulating FSS sensing signals, including statistical analysis [18], morphological image processing [19], the normalized inner-product coefficient (NIPC) [20], and the zero-mean normalized cross-correlation coefficient (ZNCC) [14]. However, most of these methods analyze the sensing parameters by calculating the correlation coefficients between the speckle patterns, resulting in limited measurement range and resolution. Deep learning-based demodulation schemes have also been proposed and proven effective through rigorous experiments [21,22,23,24,25,26]. Specifically, the speckle patterns collected from the fiber output and the corresponding sensing parameters are considered training samples and labels of the dataset, respectively, and the produced dataset is used to train the neural network. The trained model can predict the sensing parameters based only on the collected unknown samples. In 2017, E. Fujiwara et al. proposed a deep learning-based fiber specklegram sensor for evaluating force myography signals, which detects the applied force by a modular structure attached to the forearms of subjects and can estimate the actual hand configuration from the acquired optical data [21]. Test results showed an average accuracy of 89.9% for estimating finger configuration, even when a smaller number of sensors were used. In 2020, L. Yan et al. used specklegrams collected from the output of the multimode fiber to characterize the variation of mode interference induced by curvature, and employed a deep learning model to perform bending recognition according to the speckle pattern [25]. The test results show that the average accuracy of bending state recognition is 92.8% and 96.6% when the diameters of multimode fibers are 105 μm and 200 μm, respectively. In 2022, Q. Liang et al. proposed and experimentally demonstrated a learning-based fiber specklegram pressure sensor, which has a pressure resolution and recognition accuracy of 0.001 MPa and 99.97%, respectively, with the help of convolutional neural networks [24]. In addition, Q. Liang et al. used a quadrant detector instead of a camera to detect speckle images to explore the demodulation of speckle patterns in high-speed measurement scenarios. The recognition accuracies of the quadrant detector-based speckle pattern demodulation scheme were 88.89% and 99.53% when the pressure resolutions were 0.001 MPa and 0.001 MPa, respectively. Compared with conventional schemes, the learning-based demodulation scheme not only makes full use of the large amount of information carried by the fiber modes but also has good real-time performance. In addition, the measurement range depends only on the calibration range of the samples contained in the dataset.

To improve the sensitivity of sensors, it is often necessary to modify or fabricate special structures on optical fibers to enhance the interaction between light and matter. In recent years, manufacturing strategies such as electron beam lithography (EBL) [27], focused ion beam milling (FIB) [28,29], direct laser writing (DLW) [30], nano-imprinting, and laser micro-machining have been successfully applied in the fabrication of fiber optic sensors. Among them, EBL, DLW, nano-imprinting, and other techniques have limited applicability, since they typically rely on flat surfaces for manufacturing [28]. In contrast, FIB technology can easily modify or fabricate special structures on curved surfaces. In addition, FIB milling also offers advantages over other techniques, such as a simple process, no polymer coating, high accuracy, and good repeatability. At present, optical fiber sensors based on FIB milling have been successfully used to monitor the fluctuations in temperature [29] and refractive index [28]. In 2011, J. Kou et al. used the FIB method to fabricate miniaturized metal-dielectric hybrid fiber tip gratings with periodic notches for refractive index sensing, which can be used as a multichannel sensor for the simultaneous measurement of the refractive index and temperature [31]. When the period and depth of the grating are 578 nm and 650 nm, respectively, the sensitivity of the fabricated sensor is 125 nm/RIU in the refractive index range from 1.3577 to 1.3739. In 2011, J. Kou et al. used the FIB method to fabricate an all-silicon first-order fiber Bragg grating for temperature sensing, which was made by etching periodic structures on a tapered fiber probe [29]. The sensitivity of the fabricated sensor was 20 pm/°C over the temperature range of room temperature to 500 °C when the period, depth, and length of the grating were 600 nm, 200 nm, and 36.6 μm, respectively.

In this work, we propose a novel FSS based on FIB milling and neural network model demodulation for monitoring refractive index fluctuations. Specifically, an FSS with an MMF-SMF-MMF structure is first prepared by splicing a single-mode optical fiber (SMF) between two multimode optical fibers (MMF) without eccentricity. Then, a series of through channels are fabricated on the core of the SMF using FIB milling to enhance the light-matter interaction. Finally, the speckle pattern collected from the output of the MMF is used to train the neural network. The proposed scheme is experimentally characterized, and the test results show that the demodulation accuracy of the trained model can reach 99.68%, while the demodulation speed is 4.5 milliseconds per frame. In addition, the long-term quantitative results on the stability of the proposed scheme indicate that the proposed sensing system is robust. The correlation between the speckle patterns is maintained above 98% for at least 8 h, while the demodulation accuracy is consistently greater than 95% for 24 h. The proposed scheme has the advantages of easy implementation, high accuracy, and good stability, and has the potential to be a low-cost alternative to traditional sensing technology and provides an enlightening reference for using learning to solve sensing problems.

## 2. Methods

### 2.1. Design Principle

After the illumination light is coupled into the MMF, a vast amount of eigenmodes with different propagation constants are excited and propagate along the fiber, generating a speckle pattern with a highly random intensity distribution at the distal end of the MMF. Although seemingly random, the speckle pattern can be represented as a coherent superposition between the excited eigenmodes [32]:(1)$$A(x,y)=\sum_{m=0}^{M}a_{m}(x,y)\exp(j\left[\phi_{m}(x,y)\right])$$
where *M* is the number of eigenmodes excited in MMF, $a_{m}(x,y)$ is the amplitude distribution of the *m*-th mode, and $\phi_{m}(x,y)$ is the phase distribution of the *m*-th mode. Since only the intensity distribution of the speckle pattern can be obtained experimentally, the collected specklegram can be expressed as [32]:(2)$$I(x,y)={\left|A(x,y)\right|}^{2}=\sum_{n=0}^{M}\sum_{m=0}^{M}a_{m}a_{n}\exp(j\left[\phi_{m}-\phi_{n}\right])$$

It can be found that the intensity distribution of the speckle pattern depends on the interference between the eigenmodes and is extremely sensitive to phase variation. Any change in mode transmission will cause variation in the speckle pattern. When the external environment changes, it causes the phase difference and energy coupling between the excited modes to become different, leading to the speckle pattern variation. Therefore, the refractive index of the fluid can be predicted by demodulating the speckle pattern.

The scheme proposed in this work is described in Figure 1. The first step is to manufacture sensors and collect samples under different configurations to construct datasets. A neural network model is then used to learn the relationship between the speckle pattern and the refractive index of the fluid in the solution space provided by the dataset containing a large number of configurations. Finally, the trained model is employed to demodulate the unknown samples and to predict the corresponding refractive index based on the speckle pattern only. The deep learning model used in this work is the ResNet architecture [33,34], which won the ILSVRC (ImageNet Large Scale Visual Recognition Challenge) in 2015. Compared to traditional convolutional neural networks, the ResNet model demonstrates extraordinary potential in addressing gradient explosion, gradient disappearance, and degradation. The ResNet model mainly comprises a feature extraction module and a classification module. In the feature extraction module, block structures composed of convolution layers, pooling layers, and activation functions are usually applied to extract the abstract information of given samples. The classification module uses a fully connected layer to classify objects based on the extracted high-dimensional information. ResNet follows the convolutional layer (3 × 3) architecture of VGGNet (Visual Geometry Group), which is composed of a series of residual blocks. Each residual block contains two convolutional layers (the size of the convolutional layers is 3 × 3) with the same number of output channels, and each convolutional layer is followed by a batch normalization layer and a ReLU activation function. Then, the input samples skip the two convolution layers through the cross-layer data path and are added directly before the ReLU activation function. To meet such a requirement, it is necessary to ensure that the outputs of the two convolutional layers in the residual block have the same shape as the input. If the number of channels needs to be changed, an additional 1 × 1 convolutional layer needs to be introduced to transform the input into the desired configuration before doing the summation operation. Since there is a clear and obvious correspondence between the scatter pattern and the ambient refractive index in this work, ResNet18, which has a simple architecture and fast training speed, is employed as the demodulation model. ResNet18 consists of 17 convolutional layers and 1 fully connected layer.

### 2.2. Experimental Setup and Data Acquisition

The optical configuration used to collect the samples is depicted in Figure 2. A laser (Santec, Aichi, Japan, TSL-550) with a central wavelength of 1550 nm is employed as the illumination source. The light emitted from the laser is coupled into the proximal end of the FSS using a microscope objective (OBJ1, Nikon, Tokyo, Japan, CFI 20X, 20×, NA = 0.40). The sensor with the MMF-SMF-MMF structure is made by fusing the SMF (30 mm length, the diameters of the core and cladding are 8.2 μm and 125 μm, respectively) without eccentricity between two MMFs (the diameters of the core and cladding are 62.5 μm and 125 μm, respectively). Five fluid channels are etched on the SMF to enhance the interaction between light and matter. The fluid channels are carved using the FIB system (ZEISS, Oberkochen, Germany, Crossbeam 550), and the ion beam acceleration voltage and current are 30.0 kV and 100 nA, respectively. The upper panel of Figure 2 shows the scanning electron microscope (SEM, ZEISS, Oberkochen, Germany, Crossbeam 550) image of the fabricated sensor: the diameter of the fluid channels is 25 µm, the distance between adjacent fluid channels is 300 µm, and the fluid channels run through the entire SMF. The intensity of the speckle pattern emerging from the output plane of the sensor is imaged by a second microscope objective (OBJ2, Nikon, Tokyo, Japan, CFI100X, 100×, NA = 0.90) on a charge-coupled device (CCD) camera (Xenics, Suzhou, China, Bobcat 640 GigE, 16-bit, spectral range: 900 nm–1700 nm).

To learn the relationship between the speckle pattern and the refractive index of the fluid, a dataset containing different configurations needs to be constructed. Firstly, 23 different concentrations of NaCl solutions with refractive indices ranging from 1.3326 to 1.3627 were prepared. The fabricated sensor was then immersed in NaCl solutions with different refractive indices in turn, and the corresponding speckle patterns were collected at the output end of the sensor. Twenty speckle patterns were collected for each configuration, and the collection process was repeated ten times. The collected samples were cropped into a window centered on the speckle pattern and then made into a dataset containing 24 different configurations, including the original state (placed in the air) and 23 different concentrations of NaCl solutions. There are a total of 4800 images in the dataset, divided into a training set, a validation set, and a test set in a ratio of 2:1:2. To reduce time costs, the samples in the dataset were downsampled to 224 × 224 pixels. The training set provides a solution space containing different configurations where the convolutional neural network can learn the mapping between the speckle pattern and the refractive index of the fluid. The validation set is used to assess the generalization ability and error of the model during training, and to provide insight into the update of the hyperparameters. The task of the test set is to characterize the generalization ability and robustness of the trained model.

## 3. Results and Discussion

The ResNet18 model used in this work was implemented on a computer equipped with an NVIDIA RTX2060 graphics processing unit and an i7-10857H CPU. When learning the relationship between speckle pattern and refractive index using ResNet18, the solver was specified as Adam, the maximum number of epochs was 10, and the batch size was set to 20. In addition, the initial learning rate was set to 0.01 and adjusted every five epochs. To alleviate the uncertainty caused by initial value sensitivity, the transfer learning strategy is employed in the training process of the ResNet18 model, which facilitates learning ability and convergence speed. The constructed model is then trained using a homemade dataset consisting of speckle patterns.

Before training the neural network, the images contained in the dataset need to be pre-processed. In this work, the size of the speckle pattern recorded by the camera is 640 × 512 pixels. High-resolution samples do not significantly improve the accuracy of the model but instead severely slow down the convergence rate, so the collected high-resolution images cannot be directly used as input for the neural network. In this work, the collected speckle patterns are cut into specklegram-centered windows and down-sampled to 224 × 224 pixels. To visually demonstrate the variation of the speckle pattern caused by the refractive index, specklegrams corresponding to different concentrations of NaCl solution were collected and displayed in Figure 3. The upper panel of Figure 3 shows the speckle pattern corresponding to refractive indices of 1 (in the air), 1.3326, 1.3454, 1.3545 and 1.3627, respectively, while the bottom panel shows the difference between adjacent patterns. It can be found that there is a significant difference between the speckle patterns corresponding to different concentrations of NaCl solutions, which is consistent with the previous analysis and sufficient to be learned by the neural network.

Then, the ResNet18 model is used to learn the relationship between the speckle pattern and the refractive index of the fluid on the pre-processed dataset. The learning curve of the ResNet18 model is shown in Figure 4, where Figure 4a depicts the relationship between classification accuracy and epochs during training, while the loss is plotted as a function of epochs in Figure 4b. From the learning curve, it can be found that the ResNet18 model converges after four epochs, while the accuracy and training time are 99.77% and 212 s, respectively. The rapid convergence of the model proves that there are clear and significant differences between different configurations, and the learning model has fully learned the relationship between the speckle pattern and the refractive index. In addition, the model shows similar classification accuracy on both the training and validation sets, demonstrating the good generalization ability of the model. After 10 epochs, the ResNet18 model converges ultimately, and the accuracy and training time of the model are 99.93% and 530 s, respectively.

The test set is used to further quantify the generalization ability of the trained model. Unlike the training and validation sets, the test set contains samples that the model has never seen during the training process, thus allowing for a more accurate assessment of the generalization ability and the applicability of the trained model. On the test set, the trained model exhibits good real-time performance, and the classification speed of the model is 4.5 ms per frame. By convention, the confusion matrix of the trained model on the test set was calculated, as shown in Figure 5. The confusion matrix is a tool used to characterize the classification accuracy of a neural network and to visualize the deviation between the predicted and true values. The rows of the confusion matrix represent the predicted categories, while the columns represent the true categories. The elements contained in the confusion matrix indicate the number of samples classified into a particular category. From Figure 5, it can be found that most of the elements in the confusion matrix are clustered on the diagonal, indicating good agreement between the true values and the predicted values obtained using the trained model, further demonstrating the robustness and generalization ability of the model. The accuracy of the trained model on the test set is 99.68% and the resolution of the sensor is 0.0011.

The absolute classification error of the trained model on the test set was calculated to further investigate the demodulation capability of the proposed model. Figure 6 depicts the histogram of the absolute classification error of the trained model on the test set. It can be found that most of the unknown samples are classified into the correct category, and only a tiny number of samples are misidentified. Moreover, most of the errors are concentrated around the target values. The standard deviation of the absolute classification error of the model is 0.1021. This can be attributed to the fact that when the difference in ambient refractive index is relatively small and the similarity between the corresponding speckle patterns is relatively high, it can lead to model misidentification.

Vibration, air convection, and ambient temperature fluctuations introduce uncertainty in the measurement process and in the calibration process. In order to estimate the measurement uncertainty, a long-term quantification of the stability of the measurement system was performed. The fabricated sensor was immersed in a NaCl solution with a refractive index of 1.3451 and kept constant for about 8 h. A speckle pattern is collected from the output of the sensor every 20 minutes and the Pearson correlation coefficient (PCC) is used to characterize the correlation between these speckle patterns. The PCC is often used to characterize the correlation between two images; the closer the PCC is to 1, the stronger the correlation is. On the contrary, if the correlation coefficient is closer to 0, it means that the two images are not correlated with each other. The test results are shown in Figure 7a. It can be found that although the correlation between the speckle patterns decreases with time, the correlation always remains above 98% for at least 8 h, proving the stability and robustness of the measurement system. Figure 7b shows the time decay of the sensing calibration accuracy. It can be found that the demodulation accuracy and generalization ability of the model decreases with time, which is mainly affected by the stability of the light source and the decreasing correlation of the speckle pattern. However, the classification accuracy of the trained model remains above 95% within 24 h, demonstrating the robustness of the proposed scheme. In addition, in ref. [35], N. Bagley et al. demonstrate that when the fluctuations in ambient temperature are greater (temperature fluctuations range from 25 °C to 55 °C), the uncertainty introduced by temperature variations can be effectively mitigated by constructing a dataset containing more possible configurations [35].

The superiority of the scheme proposed in this work is mainly reflected in three aspects. On the one hand, the scheme proposed in this paper has high recognition accuracy and robustness. The demodulation accuracy of the trained model can reach 99.68%, and it is still close to 96% after 24 h. On the other hand, the proposed scheme has good repeatability and machining accuracy thanks to the use of FIB milling technology. The etched fluid channel enhances the light-matter interaction, which contributes to the demodulation accuracy and response speed. Finally, the proposed scheme can be extended as a general framework to more chemical and biological sensing applications. The errors in the proposed refractive index recognition scheme mainly come from two aspects. Firstly, due to the limitations of the model architecture and training strategy, the model trained in this work may not be able to thoroughly learn the evolution pattern of the speckle pattern as well as the systematic error, which leads to a gradual degradation of the demodulation accuracy. In addition, perturbations during the experimental process, such as air convection, temperature fluctuations, and mechanical vibrations, will accumulate noise in the collected specklegrams and thus degrade the accuracy of the system. A promising optimization scheme is to improve the model architecture and training strategy, such as using regularization to prevent overfitting, dynamically adjusting the learning rate to increase the convergence speed, expanding the solution space to further learn the evolution of the speckle pattern, introducing physical models to optimize the dataset, etc. Another guaranteed optimization scheme is to improve the stability of the system and minimize the system error, such as packaging the fabricated sensors, using more stable light sources and cameras, applying vibration isolation platforms, etc. In addition, etching the fluid channel on the tapered fiber would likely further improve the performance of the sensor.

## 4. Conclusions

In this work, we propose a novel optical fiber refractive index sensor based on FIB milling and deep learning model demodulation and verify both its feasibility and effectiveness through rigorous experiments. This fiber sensor with a MMF-SMF-MMF structure increases the sensitivity by milling a series of fluid channels on a single-mode fiber using FIB, and it predicts the refractive index of the fluid to be measured based only on the speckle pattern collected from the output of the MMF. Experimental results demonstrate that the proposed sensing scheme can achieve 99.68% recognition accuracy, while the demodulation speed can reach 4.5 ms per frame. In addition, long-term quantitative results of stability show that the accuracy of the sensing system is consistently above 95% for at least 24 h, demonstrating the robustness of the proposed scheme. The proposed sensing system has the advantages of simple implementation, easy fabrication, low cost, and good robustness, and it could be a promising candidate for various chemical and biological sensing applications.

## Figures and Tables

**Figure 1 nanomaterials-13-00768-f001:**
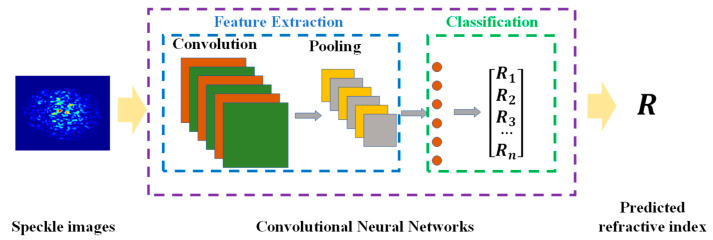
Overview of refractive index recognition scheme based on convolution neural network.

**Figure 2 nanomaterials-13-00768-f002:**
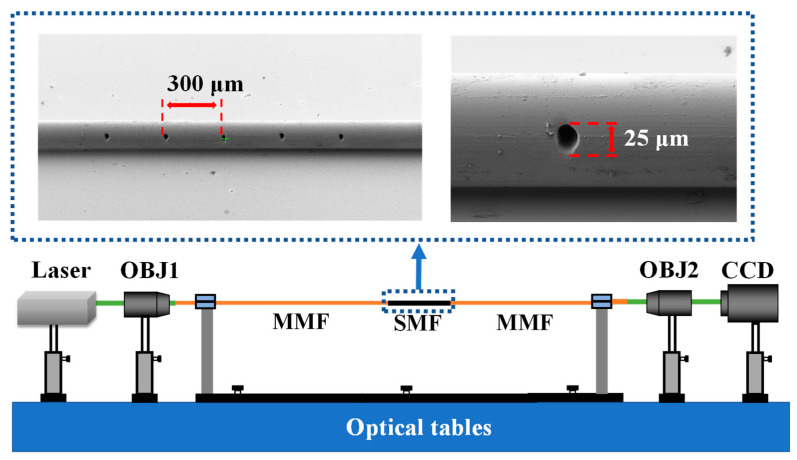
Experimental setup of the proposed fiber specklegram refractive index sensing system. CCD: charge-coupled device camera; OBJ: objective; MMF: multimode fiber; SMF: Single mode fiber.

**Figure 3 nanomaterials-13-00768-f003:**
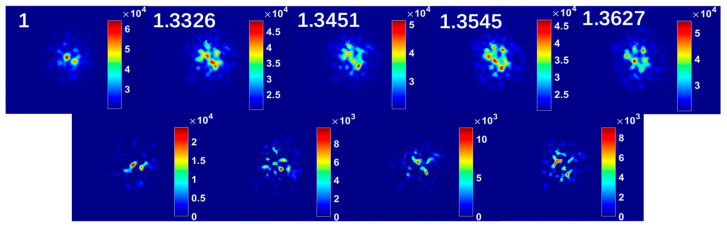
Speckle patterns corresponding to different refractive indexes and their differences between the adjacent speckle patterns.

**Figure 4 nanomaterials-13-00768-f004:**
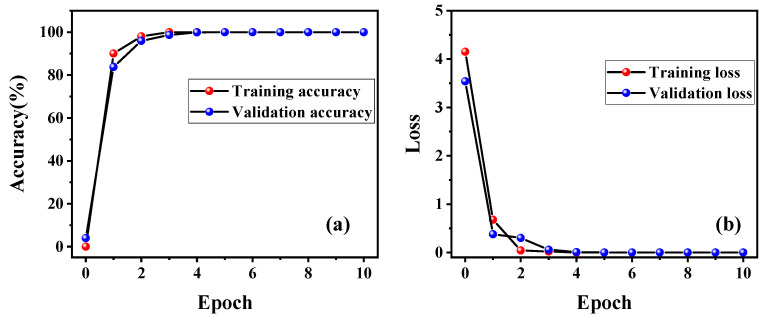
Learning curve of model based on ResNet18 architecture: (**a**) The training accuracy is plotted as a function of epochs. (**b**) The relationship between loss and epochs.

**Figure 5 nanomaterials-13-00768-f005:**
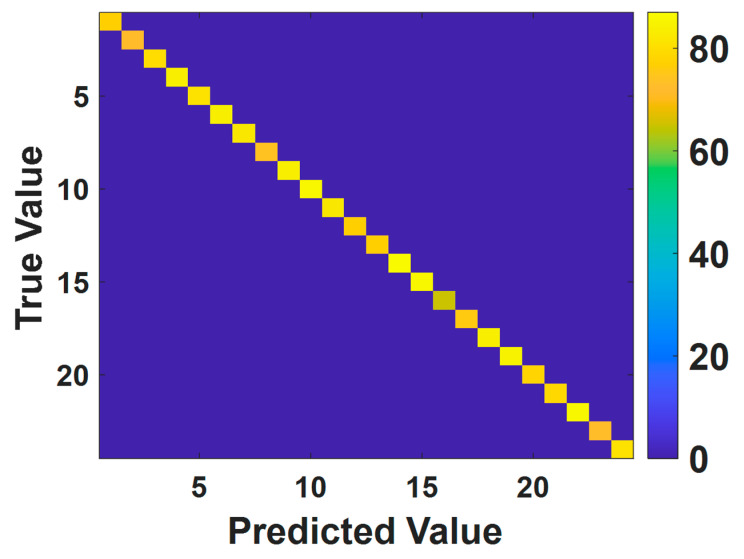
The confusion matrix of the testing set.

**Figure 6 nanomaterials-13-00768-f006:**
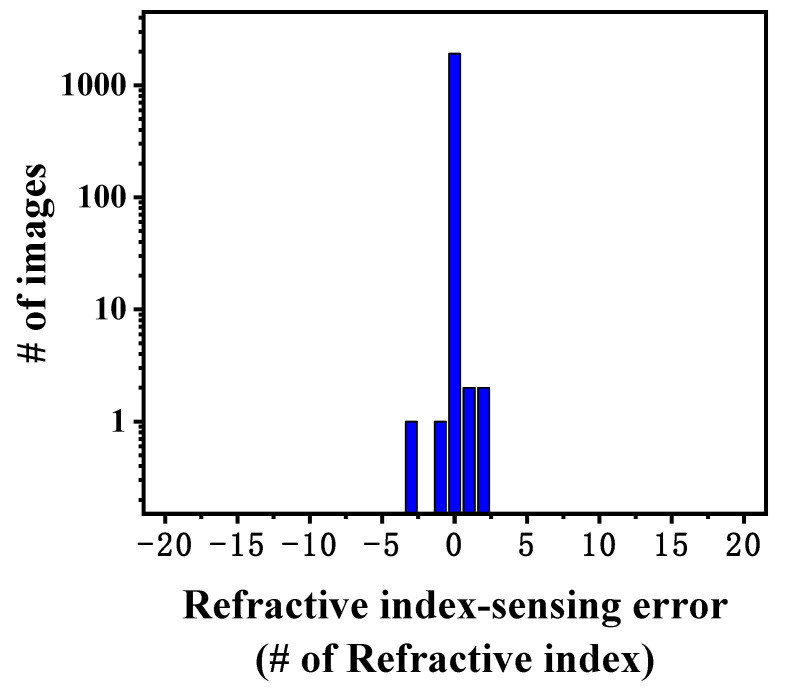
The histogram of the absolute refractive index classification error.

**Figure 7 nanomaterials-13-00768-f007:**
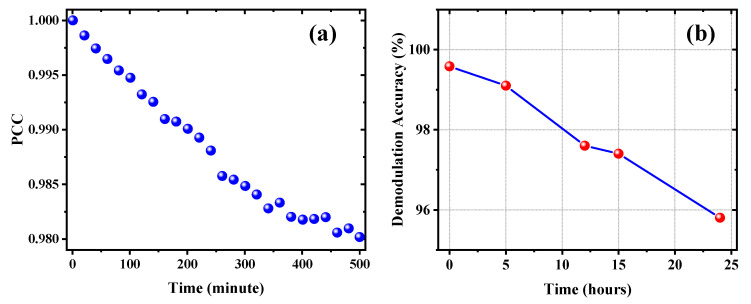
Long-term quantification of the accuracy of the sensing system: (**a**) stability test results of the sensing system; (**b**) time decay of sensing calibration accuracy.

## Data Availability

The data presented in this study are available on request from the corresponding author.

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
