# Peer review of "Demonstration of a Learning-Empowered Fiber Specklegram Sensor Based on Focused Ion Beam Milling for Refractive Index Sensing"

_nanomaterials, 2023, doi:10.3390/nano13040768_

Round 1
Reviewer 1 Report
Currently, the paper lacks some vital information. I have the following suggestions:
1) Please provide the sensitivity of the device in terms of some unit pixel/RIU or nm/RIU.
2) What is the temperature dependence on the stability and accuracy of the sensing device? The author didn't discuss this aspect.
3) I am not convinced by the literature review in the paper as almost all the references are taken from one country. Try to bring diversity to the literature and add some international papers.
4) Replace the word "fig" with "figure" in the text.
5) Provide the LoD and Quality factor of the device?
6) What is the relationship between the number of holes drilled in the SMF and the device sensitivity or performance?
7) Does the performance of the device depend on the polarization and wavelength?
Reviewer 2 Report
In this paper, authors discuss "Demonstration of a Learning-Empowered Fiber Specklegram Sensor Based on Focused Ion Beam Milling for Refractive Index Sensing". In order to enhance the interaction between light and matter, he elaborated a series of fluid channels using focused ion beam milling on the single-mode fiber. Based on the speckle patterns collected from the output end of the 16 multimode fiber, a deep learning model is used to de-15 modulate the sensing signal. For the refractive index range of 1.3326 to 1.3679, the demodulation accuracy and speed could reach 18.99.68% and 4.5 ms per frame, respectively, proving the feasibility and effectiveness of the proposed scheme. A low-cost, easy-to-implement, and simple measurement system makes the proposed sensing scheme ideal for various chemical and biological applications. The work looks interesting and worth publishing. There are a few major concerns I have
1. It would be helpful if the authors provided some transmission curve measurements or CCD images for different refractive indexes to demonstrate the feasibility of their work.
2. The author should explain and calculate the difference in results if the fiber is tapered.
3. Also, authors need to discuss the standard deviation of their results and interpreate them in their manuscripts.
Round 2
Reviewer 1 Report
I am willing to accept the paper in its current form.
Reviewer 2 Report
Comments have already been addressed by the authors. There are no further comments from me.